# Towards the definition of a solar forcing dataset for CMIP7

Bernd Funke[1], Thierry Dudok de Wit[2,3], Ilaria Ermolli[4], Margit Haberreiter[5], Doug Kinnison[6], Daniel Marsh[6,7], Hilde Nesse[8], Annika Seppälä[9], Miriam Sinnhuber[10], and Ilya Usoskin[11]

[1]Instituto de Astrofísica de Andalucía, CSIC, Granada, Spain
[2]University of Orléans, Orléans, France
[3]International Space Science Institute, Bern, Switzerland
[4]INAF Osservatorio Astronomico di Roma, Monte Porzio Catone, Italy
[5]Physical-Meteorological Observatory Davos / World Radiation Center, Davos, Switzerland
[6]National Center for Atmospheric Research, Boulder, CO 80301, USA
[7]University of Leeds, Leeds, LS2 9JT, UK
[8]Department of physics and technology, University of Bergen, Bergen, Norway
[9]Department of Physics, University of Otago, Dunedin, New Zealand
[10]Karlsruhe Institute of Technology, Karlsruhe, Germany
[11]University of Oulu, Oulu, Finland

**Correspondence:** Bernd Funke (bernd@iaa.es)

**Abstract.** The solar forcing prepared for the 6[th] round of the Coupled Model Intercomparison Project (CMIP6) has been used extensively in climate model experiments and has been tested in various intercomparison studies. Recently, an International Space Science Institute (ISSI) Working Group has been established to revisit the solar forcing recommendations, based on the lessons learned from CMIP6, and to assess new data-sets that have become available, in order to define a roadmap for building a revised and extended historical solar forcing data-set for the upcoming 7[th] round of CMIP. This paper identifies needs for improvements and outlines a strategy to address them in the planned new solar forcing dataset. Proposed major changes include the adoption of the new TSIS-1 solar reference spectrum for solar spectral irradiance and an improved description of top of the atmosphere energetic electron fluxes, as well as their reconstruction back to 1850 by means of geomagnetic proxy data. In addition, there is an urgent need to consider the proposed updates in the ozone forcing data-set in order to ensure a self-consistent solar forcing in coupled models without interactive chemistry. Regarding future solar forcing, we propose consideration of stochastic ensemble forcing scenarios, ideally in concert with other natural forcings, in order to allow for realistic projections of natural forcing uncertainties.

## 1  Introduction

Back in 2017, solar forcing recommendations for the 6[th] round of the Coupled Model Intercomparison Project (CMIP6) were provided which covered, for the first time, all relevant solar irradiance and energetic particle precipitation (EPP) contributions (Matthes et al., 2017, hereinafter referred to as M17). Since that time, new data-sets have become available, both for the solar spectral irradiance and for energetic particle fluxes in the middle and upper atmosphere. These new data-sets, if adopted, would introduce changes in the radiative forcing of climate, either directly or via their influence on atmospheric composition. The

next round of CMIP is imminent and modeling groups around the world are ensuring their models can reproduce reasonable climate states for the pre-industrial conditions as well as reproduce the historical temperature record. It is therefore essential that the forcing data-sets be revised in a timely manner.

CMIP6 brought several major improvements over prior rounds. For the first time it provided a recommendation for solar particle forcing and a comprehensive solar spectral irradiance dataset covering the full solar spectrum, including the Extreme-UV band ($10 - 121$ nm). These data-sets included a historical period with daily data from 1850 till 2015 and two different scenarios running up to 2300. However, the analysis of climate model simulations that did use the M17 data-sets also revealed some issues. For example, small changes in the shape of the solar reference spectrum (c.f. Fig. 7 of M17) induced non-negligible changes in stratospheric heating rates of up to 0.4 K day$^{-1}$ and required careful tuning of the models. The impending CMIP7 activity provides a unique opportunity to revisit these results and propose improved solar forcings.

The purpose of this perspective paper is to outline a roadmap and timeline for revising the historical solar forcing data-sets to be used in CMIP7, based on the lessons learned from CMIP6. This paper aims at 1) including the latest scientific advances made in the reconstruction of solar forcing and in the understanding of climate response, while also 2) addressing the issues that were raised during CMIP6, and 3) facilitating the practical implementation of these data-sets, both in terms of their production and their exploitation by end users. An important aspect of this work is the need for community feedback, as this will eventually help us translate these suggestions into recommendations for CMIP7.

Note that the development and documentation of updated and expanded climate forcings for CMIP7, including the solar forcing discussed here, is coordinated by the CMIP7 Climate Forcing Task Team (https://wcrp-cmip.org/cmip7-task-teams/forcings/) established by the Working Group on Coupled Models' infrastructure and CMIP panels of the World Climate Research Programme's Earth System Modelling and Observations (ESMO) project.

## 2 Solar radiative forcing

In CMIP6, solar radiative forcing consisted of total solar irradiance (TSI), the spectrally-resolved irradiance or solar spectral irradiance (SSI), and the F10.7 index for use as a proxy for solar forcing of the ionosphere/thermosphere. The spectral coverage of the SSI was 10 nm to 100 $\mu$m, with a spectral resolution that gradually increased from 1 nm to 50 nm. A new value of the average TSI during solar minimum had also been recommended, with $1360.8 \pm 0.5$ Wm$^{-2}$. The same approach is also planned for CMIP7, with identical specifications.

There are, however, two aspects to the reconstruction of solar radiative forcing which call for re-consideration: one is the definition of the reference spectrum for the quiet Sun, and the other is the definition of the variability that comes on top of it. Although TSI variability is only around 0.1% over the solar cycle, SSI variability in the UV band and at shorter wavelengths is significantly larger.

## 2.1 A new solar reference spectrum

The CMIP6 SSI forcing data-set that was recommended by M17 is an average of two timeseries from two SSI reconstruction models: Naval Research Laboratory Solar Spectral Irradiance Version 2 (NRLSSI2, Coddington et al., 2016) and Spectral And Total Irradiance REconstruction (SATIRE, Yeo et al., 2014). Both models rely on a constant, so-called quiet-Sun reference spectrum on top of which comes the solar variability. NRLSSI2 uses a composite of quiet-Sun spectra, namely the Whole Atmospheric Interval (WHI) spectrum (Woods et al., 2009) below 300 nm, the spectrum from the Atmospheric Laboratory of Applications and Science-1 (ATLAS-1) space shuttle mission (Thuillier et al., 1998) between 300 nm and 1000 nm, the spectrum from SORCE/SIM (Harder et al., 2005) between 1000 nm and 2400 nm (also used by WHI), and beyond that Kurucz's synthetic solar model atmosphere (Kurucz, 1991). SATIRE uses the WHI spectrum in the 115–2400 nm range, extended at longer wavelengths by Kurucz's atmosphere model. Ultimately, the CMIP6 quiet-Sun spectrum is the average of both reconstruction models and therefore mixes two somewhat different background spectra.

Over the recent years, a number of additional solar reference spectra based on observations have become available. First, there is the SOLAR-ISS reference spectrum by Meftah et al. (2018), which is based on the SOLAR/SOLSPEC observations (Thuillier et al., 2009) combined with the synthetic spectrum by Kurucz (1991). Second, the quiet Sun reference spectrum using the observational SSI composite by Haberreiter et al. (2017) for the annual mean of 2008 and combined with the synthetic calculations using the COde for Solar Irradiance (COSI, Haberreiter et al., 2008, 2021). Third, Coddington et al. (2021) provide a hybrid reference spectrum which is based on the latest observations from the TSIS-1 Spectral Irradiance Monitor (TSIS-SIM, Richard et al., 2020) onboard the International Space Station (ISS). According to Richard et al. (2020) the absolute uncertainty of the TSIS-1 SIM instrument is 0.2–0.5%. This value is better than the absolute uncertainty of the WHI or the ATLAS-1 spectra, which is typically >3% in the visible range. Version 2 of the TSIS-1 reference spectrum, which is an incremental update, has recently been published (Coddington et al., 2023).

A major difference between the TSIS-1 spectrum and the quiet Sun spectrum in CMIP6 is a distinct spectral shape, with the TSIS-1 spectrum showing an irradiance that is higher by 1-5% in the visible band and lower by 1-2% in the Near-IR wavelength range (between 1000 and 2000 nm), after re-normalization to the same value of the TSI. This difference is illustrated in Fig. 1, which compares the irradiance with that of TSIS-1 for specific spectral bands.

Such differences have direct implications on the climate response. For example, Jing et al. (2021) investigated the impact of the new TSIS-1 solar spectral irradiances, compared to earlier data, in NCAR CESM2 coupled climate model simulations. They found that the energy shifts between the visible and the near-infrared parts of the spectrum can trigger surface albedo feedbacks, resulting in significant differences of modeled high latitude surface temperature and sea ice coverage.

Despite its different spectral shape, we consider the TSIS-1 reference spectrum (version 2) as the most reasonable choice for future climate simulations. Indeed, the spectrum is based on the latest measurements with significantly increased accuracy compared to prior similar measurements, and has undergone a detailed validation. In addition, it has been recommended in

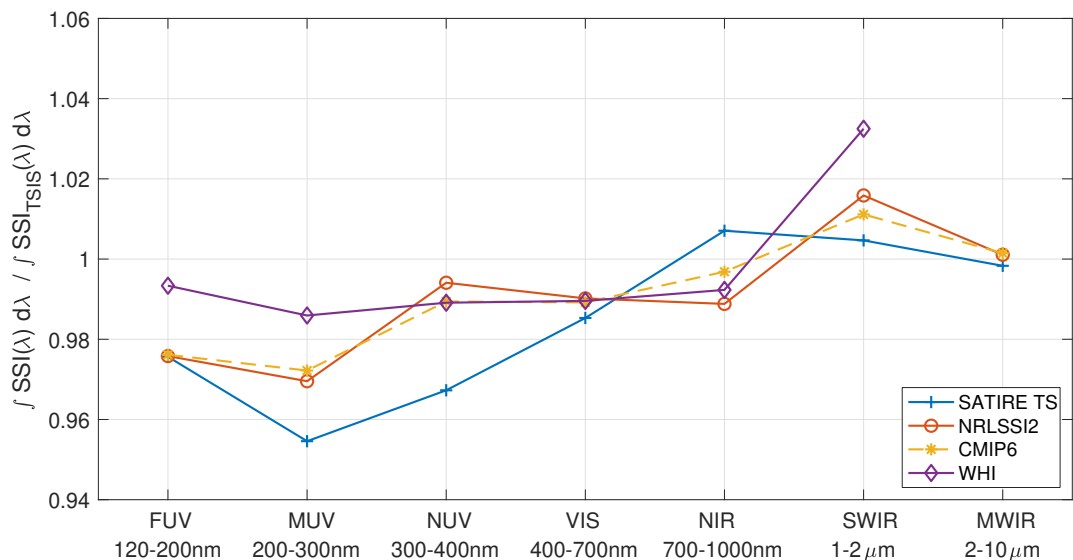

**Figure 1.** Ratio between the irradiance (in specific spectral bands) from SATIRE, NRLSSI2, CMIP6 and WHI, and that of the TSIS-1 spectrum. For CMIP6, SATIRE and NRLSSI, the reference spectrum is estimated as the mean value of the SSI for three time intervals between March 25, 2008 and April 16, 2008, which are the same as the those used for estimating the WHI reference spectrum. The spectral bands are respectively: far UV, middle UV, near UV, visible, near infrared, short wavelength infrared, and middle wavelength infrared.

March 2022 as the new reference spectrum by the Committee on Earth Observation Satellites (CEOS) Working Group on Calibration and Validation (WGCV) [1].

Finally, let us stress that in the NRLSSI2 and SATIRE models the choice of the reference spectrum is decoupled from the temporal variability of the spectral irradiance. The two are defined independently, and then added together.

## 2.2 A consistent representation of solar irradiance variability

The main challenge in making an historical solar radiative forcing dataset is the need to reconstruct SSI/TSI from proxy data for periods prior to their direct observation. Observations of the time-resolved solar spectrum from the Extreme-UV to the Near-IR became available in 2003 only, whereas direct TSI observations started in 1978 (Ermolli et al., 2013).

Nowadays many reconstructions of TSI and SSI co-exist. For instance, SATIRE derives the SSI/TSI from synthetic intensity spectra; it uses full-disk resolved filtergrams and magnetograms taken at visible wavelengths after 1974, and solar proxies such as sunspot observations before that. NRLSSI2 is more data driven as it uses measured spectra to adjust SSI variability via solar proxies. These are (since 1982) the University of Bremen Mg II measurement composite and the areas and locations of sunspots as reported by the USAF SOON sites. For sunspot region information prior to 1982 the Greenwich Observatory observations are used.

---

[1]https://calvalportal.ceos.org/tsis-1-hsrs

Besides the NRLSSI2 and SATIRE reconstructions, there are additional SSI reconstructions available. It should be noted that all reconstruction approaches, including the NRLSSI2 and SATIRE, are based on the assumptions that the irradiance variations are caused by the changing magnetic features on the surface of the Sun, but differ in the implementation of those changes. Egorova et al. (2018) use the Code for the High spectral ResolutiOn recoNstructiOn of Solar irradiance (CHRONOS). The CHRONOS model is an update of the reconstruction approach by Shapiro et al. (2011) with a revised method to derive the varying contributions of the quiet Sun, faculae, sunspot umbra and sunspot penumbra and the combined spectra. Different versions of the CHRONOS reconstruction exist, which are based on different input parameters to derive the long-term evolution of the quiet Sun irradiance, for a comparison with NRLSSI and SATIRE see Yeo et al. (2020a, Fig. 1b). For discussion of further irradiance reconstruction models we refer to the reviews by e.g., Ermolli et al. (2013) and Chatzistergos et al. (2023).

In CMIP6, we selected the only two models that could reconstruct solar irradiance over the whole period considered and that had been studied in detail. These were SATIRE and NRLSSI2. Both use various solar inputs (magnetograms, sunspot number, etc.) but differ in the way these translate into SSI/TSI variability. These differences, along with the use of different versions of proxy datasets, have led to systematic discrepancies in solar cycle amplitudes and secular trends. These have been the subject of much debate without the community reaching a consensus on the most appropriate model for climate simulations. Both models come with uncertainty estimates, but because they are based on different metrics, these estimates cannot be meaningfully compared in a quantitative way. For these reasons, it was decided for CMIP6 to average the two reconstructions without favoring either model. For CMIP5, only the NRLSSI (Lean, 2000) model was used.

While the averaging applied for CMIP6 was considered as the most sensible choice, given the available information, it has also received criticism. One problem arises from the different reference spectra that are used by both models (see Sec. 2.1), whose averaging leads to yet another spectrum of the composite. Another objection comes from the different trends that arise during the space era (after the 1980s) when SATIRE produces a stronger downward trend in the SSI observed at solar minimum when compared to NRLSSI2 and to the most recent measured TSI.

For CMIP7 the objective is to revisit these choices in the light of recent developments made by both model teams, and to find a pragmatic solution that would provide the best solar input for climate models.

In the meantime, no community consensus has been reached regarding the relative accuracy of the models. Both are continuously being improved and the agreement between them tends to improve (Lean et al., 2020). Solar surface magnetism has been confirmed to be the main driver for SSI variations (Yeo et al., 2017) and so growing attention has been given to its role, which is crucial for constraining the SSI/TSI during periods of very low solar activity, such as during the Maunder minimum (Yeo et al., 2020b; Krivova et al., 2021; Wang and Lean, 2021). New data sources are gradually becoming available such as full-disk resolved solar images taken in the Ca II K line since 1892, which provide new insight into the long-term evolution surface magnetism (Chatzistergos et al., 2022). At this stage, these new data are still mostly used for validation purposes.

Another aspect to be considered for CMIP7 is the consistency of the recommended solar forcing with that to be used for paleoclimatic reconstructions in PMIP5. At the time of writing the latter are not yet known. The recommended solar forcing for PMIP4 was based on SATIRE-M (Jungclaus et al., 2017), which was therefore not fully consistent with the one used for CMIP6.

Another issue is the production of the SSI/TSI dataset for future scenarios. As for CMIP6, we are planning to provide a set of forcing scenarios with daily values up to 2300. These will be produced from one single solar input (the sunspot number or the group number) similarly to the way historical reconstructions of the SSI/TSI are made before solar images or magnetograms became available. For internal consistency it would be preferable to use the same solar proxies, and the same version in both models. Unfortunately, different versions of the sunspot number record coexist (Clette et al., 2023), which has led to small but significant differences in historical solar forcing (Kopp et al., 2016). Finally, there are practical considerations, such as the process for building historical and future forcings, which should be flexible enough to allow for operationalization and regular updates, with a short latency.

What is the best solar irradiance forcing data-set for CMIP7? In the absence of a community consensus on the models, and by lack of comparable uncertainty estimates, the most reasonable choice would be to again average the latest versions of the two SSI/TSI models, namely SATIRE (including its recent improvements) and NRLSSI3 (or possibly NRLSSI4, which is in preparation). However, to avoid some of the problems that were encountered in CMIP6, a meaningful average requires that both models use the same reference spectrum (see the suggestion made in Sec. 2.1) and are driven by the same solar proxies, namely either sunspot number or group number. Furthermore, a consistent treatment of TSI/SSI variability in CMIP7 historical and PMIP time periods should be considered.

## 3  Energetic particle forcing

Energetic particle forcing for CMIP6 was provided in terms of atmospheric ionization rates for magnetospheric mid/medium-energy electrons (MEE), solar energetic particles (SEPs), and Galactic Cosmic Rays (GCR), as well as geomagnetic proxies (i.e., the Ap and Kp index). In addition, to capture the effects of polar winter descent of EEP- generated $NO_x$ (EPP-$NO_x$) in chemistry climate models that have an upper lid in the mesosphere (i.e., below the EPP source region), recommendations for the implementation of an odd nitrogen upper-boundary condition were provided.

Recent intercomparison studies have shown a systematic underestimation of the CMIP6 MEE ionization rates compared to other data-sets (Nesse Tyssøy et al., 2019; Clilverd et al., 2020; Mironova et al., 2019; Nesse Tyssøy et al., 2022), leading to a significant underestimation of the atmospheric response in the middle and upper mesosphere (Pettit et al., 2019; Szeląg et al., 2022). This has been attributed to a deficient description of the top of the atmosphere particle fluxes and to the Ap based reconstruction approach which does not account for the dynamics of precipitation during geomagnetic storms. Moreover, the CMIP MEE precipitations are developed based on averaged flux responses which might dampen the overall precipitating flux variability both on daily and decadal scales. These three aspects should be considered in the preparation of the solar forcing for CMIP7 and are discussed in more detail in Sections 3.1 and 3.2. Aside from this, only minor updates with respect to M17, discussed in Sec. 3.3, are proposed for CMIP7 energetic particle forcing.

### 3.1 Improved estimates of the top of the atmosphere MEE fluxes

Mid-energy electron precipitation fluxes are derived from the MEPED/POES instruments which provide observations in three energy bins ($\geq$30 keV, $\geq$100 keV, and $\geq$300 keV) and in two perpendicular viewing angles (Evans and Greer, 2000). For CMIP6, electron fluxes were extracted using data from only the 0° telescope (van de Kamp et al., 2016). The low bias of the fluxes used in CMIP6 has been primarily attributed to an underestimation of the loss cone when using only the 0° telescope. Data-sets based on an estimate of the loss cone combining the 0° and 90° telescopes and using daily observations provide higher fluxes (Nesse Tyssøy et al., 2019; Nesse Tyssøy et al., 2022) and lead to a stronger atmospheric response (Sinnhuber et al., 2022; Pettit et al., 2021). Therefore, for CMIP7 it is proposed to improve estimates of precipitating fluxes by using data from both telescopes, e.g., based on the approach of Nesse Tyssøy et al. (2016) on the new homogeneous composite developed by (Asikainen and Ruopsa, 2019) and (Asikainen, 2019) which will enable the estimate of precipitating fluxes over the full observation period from 1979 to present day in the energy range 30–1000 keV. The long observation period covering multiple solar cycles allows for a better foundation and validation of the MEE parametrization.

### 3.2 Refined reconstruction of MEE fluxes

As a response to the underestimation of the MEE fluxes in M17, updated particle flux observations as outlined in the previous section should be used to construct an updated precipitation model. Consistent with M17, we propose following the theoretical framework of van de Kamp et al. (2016) for parameterising the fluxes on L-shells in terms of geomagnetic index, however based on estimated electron fluxes using data from both MEPED/POES telescopes (see above). At this stage, no need for including the magnetic local time (MLT) dependency in the fluxes (van de Kamp et al., 2018) has been identified (Verronen et al., 2020).

Further developments to overcome the current deficiencies in the atmospheric impact could come from: 1) using an alternative geomagnetic index to Ap (e.g., the aa-index could be used directly to reconstruct the long-term data-set), 2) incorporating a lagged or an accumulated response to better represent the temporal evolution of geomagnetic storms, and 3) using a piece-wise energy spectra power law for extracting spectra in the range 30–1000 keV, rather than the single power law approach in M17. The motivation of the latter arises from seeking improvements for the fluxes in the high energy tail of the spectrum, which in M17 were likely underestimated. Separation of the spectral fits by energy range could then further allow for a delayed impact ($\sim$2 day delay) of high energy electrons, in a manner consistent with what is seen in observations (Nesse Tyssøy et al., 2021; Salice et al., 2023). Finally, if the validation reveals that the dependent variable (aa or Ap) has a wide range of possible flux responses where an average representation dampen the overall precipitating flux variability, implementing a stochastic solar cycle dependent element should be considered.

### 3.3 Further minor updates

– The atmospheric ionization rates from MEE precipitation in M17 were calculated using the formulation of Fang et al. (2010), which is accurate over the energies up to 1 MeV but does not consider the secondary Bremsstrahlung peak

at lower altitudes. A new parameterization for this calculation has been formulated for the high-energy tail from 100–1000 $keV$ considering Bremsstrahlung by Xu et al. (2021), which could be used to replace (or extend) the Fang et al. (2010) parameterizations. However, the impact on atmospheric composition of including this is likely small.

– Since M17 we are aware of no studies that have highlighted significant deficiencies in the specification of solar energetic particle (SEP) fluxes. SEP fluxes are derived from GOES satellite observations in the energy range from a few $MeV$ to 100 $MeV$, and are extrapolated to 300 $MeV$. This yields good results in the upper stratosphere and mesosphere above 35–40 km, the altitude region most affected by solar proton events (e.g., Jackman et al., 2001; Funke et al., 2011). However, in rare events with a harder spectrum, this can lead to an underestimation of the SEP impact below this altitude

(Jia et al., 2020). Therefore, the energy range should be extended from 100 MeV used in M17, to 400 $MeV$ using a recent re-calibration of the GOES detectors (Raukunen et al., 2022). This would account for the previously missing contribution of SEPs to stratospheric chemistry in the vertical range of $\sim$20–40 km. Ionization rates associated with the lower-energy part of the energetic particle spectrum can still be calculated as in M17 using the analytical approach by Jackman et al. (1980). This approach, however, cannot be applied to high-energy ($>$100 $MeV$) protons which initiate the atmospheric

nucleonic cascade and can penetrate to the lower atmosphere. For that, we propose an approach based on the ionization yield functions precomputed with a physics-based model, based on Monte Carlo simulations of the atmospheric cascade, CRAC:CRII (Usoskin and Kovaltsov, 2006; Usoskin et al., 2010, 2011; Väisänen et al., 2023). The long-term dataset covering the historical period from 1850 to 1962 can be reconstructed in the same stochastic manner as in M17.

  – Galactic Cosmic Rays (GCR) are proposed to be treated in a similar way to M17, i.e., by means of the force-field

approximation parameterized via the modulation potential $\Phi$. However, $\Phi$ should be obtained from the ground-based neutron monitor data-set that begins in 1951 (Usoskin et al., 2005; Usoskin et al., 2017) and is continuously updated at https://cosmicrays.oulu.fi/phi/phi.html. For the historical period before 1951, the $\Phi$ timeseries can be based upon the solar open-flux model of Krivova et al. (2021).

  – Geomagnetic shielding affects the spatial distribution of atmospheric ionization by GCRs, SEPs, and EEP. For CMIP7,

it is proposed to follow the approach implemented by M17 using the International Geomagnetic Reference Field (IGRF) model truncated to the eccentric tilted dipole component (the first 8 Gaussian coefficients) which is known to adequately globally represent the realistic field for the cosmic-ray shielding (Nevalainen et al., 2013). The newest version of the IGRF, the thirteenth generation model (Alken et al., 2021) is recommended to be used.

  – The main impact of energetic particle precipitation on the composition, independent on the particle source, is by the

formation of $NO_x$ (N, NO) and $HO_x$ (H, OH) by atmospheric ionization. This is implemented in chemistry-climate models (CCMs) using simple parameterizations first outlined by Porter et al. (1976) and Solomon et al. (1981), or by including the complex D-region ion chemistry (e.g., Verronen et al., 2016). The simple parameterization approach has been recommended for CMIP6 (M17). It yields too low $NO_x$ formation in the lower thermosphere (Nieder et al., 2014), but has been shown to perform well throughout the middle atmosphere below $\sim$80 km altitude in many studies. An

altitude parameterization of the $NO_x$ formation similar as for $HO_x$ could be constructed based on (Nieder et al., 2014) which, however, would only have significant implications for models extending higher than 1 Pa.

– For those CCMs with an upper lid in the mesosphere, an odd nitrogen upper-boundary condition (UBC) is required, accounting for EPP productions higher up. M17 recommended the use of the UBC model described in Funke et al. (2016), which is based on MIPAS observations taken during the 2002–2012 period. It is planned to maintain the same
approach for CMIP7, however, an extended validation of the UBC model with more recent NO observations (and possible update if required) should be considered. In addition, the use of the aa index instead of the Ap index (similar as for the MEE reconstruction) for driving the UBC model should be explored.

## 4   Uncertainty quantification

One of the requests made after the delivery of the CMIP6 dataset was the production of uncertainties, especially regarding the
solar irradiance dataset. Although the SATIRE and NRLSSI2 irradiance models come with some uncertainty estimates, turning these into complete uncertainties (at all wavelengths, for all times) that can be meaningfully compared, is difficult. In addition, such uncertainties should also distinguish long-term stability and short-term errors, which are usually referred to as precision.

For the SOLID irradiance dataset (Haberreiter et al., 2017) these two types of uncertainties were estimated directly from the data, thereby providing a homogeneous ensemble that enabled a comparison of the different models. A similar approach
should be feasible for CMIP7 for determining short-term errors. However, the estimation of the long-term stability is much more challenging. Different approaches will be explored to determine whether they can be provided at all.

## 5   Consistency of ozone forcing data-sets with solar input

An updated CMIP7 SSI input for climate models with interactive chemistry is expected to result in ozone changes over the 11-year solar cycle similar to those produced by CCMs in CMIP6 (e.g., Maycock et al., 2018). Ozone changes between solar
maxima and minima (i.e., per 130 SFU units, where 1 SFU = $10^{-22}$ $Wm^{-2}Hz^{-1}$) were approximately 2% in the tropical mid-stratosphere. For CMIP7, the historical SSI forcing will be extended through 2022 which will be important for near real time studies of both chemistry and climate impacts. Moreover, the planned transition to a new solar reference spectrum (see Sec. 2.1) implies significant changes in the spectral shape, potentially resulting in a modified climatological ozone field in the upper stratosphere and mesosphere.

Further, the CMIP6 ozone forcing data-set lacked a realistic representation of polar EPP-induced ozone impacts. This data-set was produced as a weighted composite of two CCMs, whereby one CCM (with a stronger weight in the upper stratosphere and mesosphere) did not consider EPP, while the other CCM underestimated the EPP-induced $NO_y$ perturbation in the polar stratosphere (Szeląg et al., 2022), which resulted in an underestimate of the polar ozone loss and subsequent feedback on temperature and dynamics. This study suggested that part of this discrepancy in $NO_y$ was due to an underestimation of EEP-
$NO_x$ from the MEE forcing data-set. As discussed above, the MEE forcing for the CMIP7 EPP-$NO_x$ may be 2-10 times larger

(see section 3.2). This will significantly increase the impact of particle precipitation on stratospheric ozone, at least in the upper stratosphere, making the consideration of EPP-induced variability in the ozone forcing data-set even more relevant. It would also result in a better agreement with observational estimates of the EPP impact on ozone, which indicate a 15% ozone reduction on average and solar cycle variations of about the same magnitude (Damiani et al., 2016).

As for CMIP6, ozone data-sets using CMIP7 forcings for coupled climate models with non-interactive chemistry will be supplied from models with interactive chemistry. The solar forcing influence should be just one part of the overall ozone variability that needs to be updated consistently (e.g., together with volcanic forcing and EESC). This effort will be coordinated by the CMIP7 Climate Forcing Task Team.

# 6    Release timeline

The release of a preliminary historical solar forcing data-set (beta version) is already planned for early 2024, in order to facilitate early model tuning efforts and ozone forcing generation, as well as a thorough validation of the data-set before its final release. The latter is planned for early 2025, after consideration of community feedback on the preliminary version of the historical forcing and inclusion of future scenarios. A more general overview of the timeline for generation of all CMIP7 forcings is provided by Durack et al. (2023).

# 7    Looking forward

The definition of a strategy for the generation of future solar forcing scenarios is still pending. This issue deserves further discussion in order to reach a community consensus on how to deal with projected natural forcing uncertainties. CMIP5 climate projections were based on a stationary Sun scenario (i.e., repetition of solar cycle 23). In CMIP6, this was replaced by a more plausible scenario for future solar activity, exhibiting variability at all timescales (daily to centennial) in accordance

with the Sun's past behavior. The motivation for this decision relied in the sensitivity of the response of a nonlinear (climate) system to the magnitude of the forcing variability. However, given the difficulty of predicting solar activity even one cycle ahead, it is clear that both approaches are subject to significant uncertainties. Even if some quasi-harmonic components of the solar forcing (those related to the Schwabe cycle with a periodicity of approximately 11 years) may provide some degree of predictability, other components such as sporadic solar proton events, exhibit a predominantly stochastic behavior. Associated

uncertainties may interfere with the emergence of anthropogenic signals. For instance, the date of ozone hole recovery may be under- or overestimated due to interannual to decadal variability in composition resulting from solar variability. This issue becomes even more important for the volcanic forcing, where it is unlikely that sporadic sulfate injections yield a modeled atmosphere with the same variability as one where a multi-decadal mean sulfate distribution is specified.

What is the best solution for specifying future natural forcing? None of the approaches chosen so far (steady-state vs. a

single transient scenario) is an optimal solution. Only the use of stochastic ensemble forcing scenarios would ensure a realistic quantification of the impact of natural forcing uncertainties, and thus ultimately increase confidence in climate projections.

Regarding the future solar forcing, such an ensemble could be constructed from a set of plausible evolutions of the solar activity level, i.e., considering different solar cycle lengths, amplitudes, and distribution of impulsive events like solar proton events. However, this approach would come at a cost in terms of computational resources. In summary, a debate on the strategy for accounting for future natural forcing uncertainties needs to be initiated in a broader community and should not be limited to solar forcing alone.

*Code and data availability.* No software packages were used in this article. The CMIP6 solar forcing data-set discussed in this work can be obtained from the input4MIPS repository (doi:10.22033/ESGF/input4MIPs.1122).

*Author contributions.* All authors have contributed to the writing of the manuscript.

*Competing interests.* The authors declare that they have no competing interests.

*Acknowledgements.* The authors are grateful for the International Space Science Institute for supporting the *Solar Forcings for CMIP7* working group. They also gratefully acknowledge valuable discussions and inputs from the following persons: Timo Asikainen, Stefan Bender, Mark Clilverd, Odele Coddington, Serena Criscuoli, Natalie Krivova, Judith Lean, Joshua Pettit, Erik Richard, Craig Rodger, Max van de Kamp, Pekka Verronen, and Jan Maik Wissing. TD acknowledges support from CNES. BF acknowledges support by MCIU under project PID2019-110689RB-I00/AEI/10.13039/501100011033. MH acknowledges support by Karbacher-Fonds. IU acknowledges partial support from the Academy of Finland (projects ESPERA, Grant 321882). The National Center for Atmospheric Research is a major facility sponsored by the National Science Foundation under Cooperative Agreement No. 1852977. This work was initiated in the frame of the WCRP/SPARC SOLARIS-HEPPA activity.

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
