# Peer review of "Towards the definition of a solar forcing dataset for CMIP7"

_Geoscientific Model Development, 2023_

## Author Response (AR1)

**Responses to the referee and community comments on gmd-2023-100.**

We thank all reviewers and members of the community for their valuable comments, questions and suggestions. Please find below our detailed point-by-point replies (in blue color) to the comments, which we hope have addressed all satisfactorily, as well as the actions taken on the manuscript.

**Reply to Anonymous Referee #1**

We would like to thank Referee #1 for the thoughtful and constructive suggestions and comments, which will certainly help to improve our manuscript and to consolidate the solar forcing generation for CMIP7.

In their manuscript, the authors make suggestions for the construction of a solar forcing dataset for use in the upcoming phase 7 of the Coupled Model Intercomparison Project (CMIP). The final sentence of the Introduction nicely summarizes what I see as the main point of this manuscript: "An important aspect of this work is the need for community feedback, as this will eventually help us translate these suggestions into recommendations for CMIP7." I find the publication of such suggestions very timely and useful, and I would very much like to see this manuscript be published quickly because, as the authors mention, the solar forcing dataset is a prerequisite for the construction of an ozone dataset to be used by models without interactive ozone chemistry, and it would be very useful to coordinate with the PMIP community concerning the construction of solar forcing data for the deeper past. For all modeling groups that want to participate in CMIP7 the preparation is getting more difficult the later the necessary input datasets will be available. I have only minor suggestions which I'd like the authors to consider before I can recommend publication of the manuscript. Two of them are more general, the others are listed below ordered by their occurrence in the text.

First, it would be very useful to clarify the role and mandate (if existing) of the group of authors for the construction of the envisaged CMIP7 solar forcing dataset.

Reply: Most of the authors (seven out of ten) were involved in the generation of the solar forcing dataset for CMIP6. The construction of the envisaged CMIP7 solar forcing dataset is endorsed by the CMIP7 Climate forcing task group (see comment below) of which the first author of this manuscript is a member. The CMIP7 solar forcing preparation is coordinated by the first author of this manuscript, who also co-led the CMIP6 solar forcing development.

Concerning ozone, in Chapter 4 it is said that "This effort will be coordinated by the CMIP7 Climate Forcing Task Team." What is this team, and which role does it play concerning solar forcing?

Reply: The World Climate Research Programme (WCRP) Earth System Modelling and Observations (ESMO) project, through its Working Group on Coupled Models' (WGCM) CMIP panel and WGCM infrastructure panel (WIP), has established a number of Task Teams to support the design, scope, and definition of the next phase of CMIP and evolution of CMIP infrastructure and future operationalization. One of these Task Teams is the Climate Forcing Task Team (https://wcrp-cmip.org/cmip7-task-teams/forcings/, see also CMIP Annual report 20202-2023, doi:10.5281/zenodo.8101810), whose core goals is, among others, to work with teams (as ours) to identify, develop, document and deliver an updated and expanded forcing collection to near real time for CMIP7.

We have added the following sentence in the Introduction: "Note that the development and documentation of updated and expanded climate forcings for CMIP7, including the solar forcing discussed here, is coordinated by the CMIP7 Climate Forcing Task Team (wcrp-cmip.org/cmip7-task-teams/forcings/) established by the Working Group on Coupled Models' infrastructure and CMIP panels of the World Climate Research Programme's Earth System Modelling and Observations (ESMO) project."

It would also be very useful to mention where the team of authors will become active themselves in the construction and where they rely on input from other groups. For example, from my reading of Chapter 2 I got the impression that a reference quiet-Sun spectrum is available, but for the irradiance variability the authors may depend on input from the SATIRE and/or NRLSSI groups. Is there any commitment from these groups or at least an established forum for the potentially necessary coordination?

Reply: Both SATIRE and NRLSSI teams are committed to providing their SSI reconstructions.

It would also be useful to more clearly address a coordination with the PMIP group. Lines 106 ff. sound to me like the groups of authors will wait for a recommendation of the PMIP group. I think it would be very useful to search for a consensus proactively.

Reply: We completely agree with this comment and have already approached the PMIP group.

Concerning particle forcing it sounds to me that the group of authors may have all the tools and data available which are necessary to produce the forcing data themselves. Is this correct or not? Please spell this out.

Reply: The generation of the particle forcing is coordinated by authors of this manuscript and involves a larger group of scientists, all of them having expressed their commitment to contribute to this effort.

Second, I think that for climate modellers, and if comments from them are intended, it would be very useful to more comprehensively discuss the impacts of the choices which have to be made for constructing the solar forcing dataset. In the introduction it is said that "new data-sets, if adopted, would introduce changes in the radiative forcing of climate, either directly or via their influence on atmospheric composition". Why not summarize and, wherever possible, quantify the major effects somewhere in the manuscript?

Reply: We have extended the discussion about the impacts of the proposed forcing updates in Sec. 2.1, including a figure of the expected SSI changes in different wavelength regions.

The definition of future solar forcing is left, to my reading, relatively open in the final section of the manuscript. As also some future forcing is needed, I'd like to see a more specific suggestion also for this period.

Reply: We intentionally put the focus of this paper on the historical period, as the timeline for its construction is tight and its availability is critical for the preparation of CMIP7 climate model simulations (as also noted by this reviewer). The timeline for the future forcing is more relaxed and, more importantly, a general community discussion on the treatment of natural forcing variability in CMIP7 projections is required before starting with its production. To initiate such a discussion is the main purpose of the final section of our manuscript.

It may be useful to consider lessons learned from earlier CMIP phases. Sedlacek et al. (Earth and Space Sciences, 2023; probably published after the submission of this manuscript) come to the relatively strong conclusion that it may not be necessary to provide different future scenarios: "Our results indicate that low amplitude solar forcings such as the EXT CMIP6 or similar are not worthwhile considering during the next CMIP type of activities." There may be other opinions, but I think this result should be discussed.

Reply: We agree with the findings of Sedlacek et al. with respect to the impact of different secular trends in solar activity. These findings are also in line with previous assessments. However, this study (and previous studies) did not assess the impact of the uncertain solar forcing evolution on shorter scales (e.g., related to solar cycle progression and impulsive events), which might introduce important natural forcing uncertainties on the annual to decadal timescales. This impact cannot be assessed by means of CMIP6 future scenarios since both REF and EXT scenarios were based on one single projection of historical cycles into the future (however differently scaled).

On the other hand I could imagine that the relatively large energy shifts between the visible and near-infrared parts of the solar spectrum in the new TSIS compared to earlier data, as reported by the authors, may have a non-negligible impact. Are estimates of the impact already possible?

Reply: Jing et al. (2021) investigated the impact of the new TSIS-1 solar spectral irradiances, compared to earlier data, in NCAR CESM2 coupled climate model simulations. They found that the energy shifts between the visible and the near-infrared parts of the spectrum can trigger surface albedo feedbacks, resulting in significant differences of modelled high latitude surface temperature and sea ice coverage. A brief discussion along these lines and a reference to Jing et al. (2021) paper have been added to the manuscript.

Jing, X., X. Huang, X. Chen, D. L. Wu, P. Pilewskie, O. Coddington, and E. Richard, 2021: Direct Influence of Solar Spectral Irradiance on the High-Latitude Surface Climate. J. Climate, 34, 4145–4158, https://doi.org/10.1175/JCLI-D-20-0743.1.

L25: "However, the analysis of climate model simulations that did use the M17 data-sets also revealed some issues. Small changes in the shape of the solar reference spectrum, for example, impacted climate simulations and required careful tuning of the models." In the spirit of what I wrote above: Here it would be very useful to be more specific and provide references if possible. What were the issues? Which small changes caused which impacts?

Reply: Here we refer to the CMIP6 versus CMIP5 SSI differences as shown in Fig. 7 of Matthes et al. (2017). Such differences were shown to produce changes in stratospheric heating rates of up to 0.4 K/day and associated temperature changes of up to 1.5K (see Fig. 6 of Matthes et al., 2017). This information has ben added.

L52: "SATIRE uses the WHI spectrum, where available …" For which wavelengths?

Reply: The WHI spectrum is used in the 115-2400 nm range in SATIRE. This information has been added.

L85: "NRLSSI2 is more data driven as it relies on solar proxies only." I may not understand which data are referred to, here, and what "more data driven" actually means.

Reply: Our statement is admittedly misleading as NRLSSI2 uses measured solar irradiance spectra to adjust SSI variability via solar proxies. Proxies used are (since 1982) the University of Bremen Mg II measurement composite and the areas and locations of sunspots as reported by the USAF SOON sites. For sunspot region information prior to 1982 the Greenwich Observatory observations are used. This has been clarified in the revised version.

L106: "Other aspects …" That's only one aspect, right?

Reply: Yes. We have changed the text to "Another aspect to be considered…".

L113: "… in both models. Unfortunately, different versions coexist …" Which models? I guess SATIRE and NRLSSI but as they are not mentioned in the lines above it might be good to say that again. And please be more specific concerning the different versions.

Reply: We here refer to the sunspot number or group number. This has been added in the revised version.

L115: Why should the process "be flexible enough to allow for yearly updates"? I do see the appeal of a flexible tool, but in view of CMIP yearly updates wouldn't be necessary.

Reply: The usage of CMIP forcing datasets in the past was not restricted to CMIP model simulations and a broad application range is also expected for CMIP7. Regularly updated forcings would be particularly beneficial for annual to decadal climate prediction as envisaged e.g. in the frame of WCRP's Explaining and Predicting Earth System Change (EPESC) Lighthouse activity. A user demand for regularly updated forcings has been identified and ways forward are currently discussed within CMIP7 Climate Forcing Task Team, see doi:10.5281/zenodo.8046147.

L130: "leading to a significant underestimation of the atmospheric response in the middle and upper mesosphere (Smith-Johnsen et al., 2018; Sinnhuber et al., 2022; Szelag et al., 2022)" and L144 "lead to a stronger atmospheric response (Sinnhuber et al., 2022; Pettit et al., 2021)" I only checked the Sinnhuber et al. paper, but their conclusions don't seem to fit well to these statements, they, e.g., write: "In the high-latitude upper mesosphere and lower thermosphere above 80 km altitude, multi-model mean results of NO using different MEE ionization rate data-sets are very similar", and "In the high-latitude mesosphere below 85 […] it is not possible to provide a robust estimate as to which of the ionization rate data-sets perform best", and "All three observational data-sets agree on a significant NO enhancement during and after the geomagnetic storm at and below 70 km altitude." So I suggest to summarize the potential impacts of an improved dataset on the atmospheric in a more nuanced way.

Reply: We thank the reviewer for pointing this out. The reference to Sinnhuber et al. (2022) and Smith-Johnsen et al. (2018) in L130 can be misleading; Sinnhuber et al. (2022) did indeed find a significant difference between multi-model mean results using different ionization rates, but unfortunately in this special case, the differences between three observational data-sets used to assess these model results were even larger. Smith-Johnsen et al. (2018) show a better agreement with observations if medium-energy electrons (MEE) are taken into consideration compared to considering purely auroral forcing, but still greatly underestimate the impact in 90-110 km with their MEE data-set using only the zero-degree channel of POES; however, they did not use the CMIP6 MEE data-set, but the BCSS-FRES data. Szelag et al. (2022) did indeed use the CMIP6 MEE forcing, and found that their EPP-NOy was up to an order of magnitude lower than observations based on a 48-year transient simulation. An underestimation of EPP-

NOy is also shown for a model experiment using the CMIP6 MEE data-set compared to satellite observations in Pettit et al. (2019) for the austral winter 2003. The authors also highlight a significant improvement when both MEPED telescopes are considered. We therefore have changed L130 to read "… leading to a significant underestimation of the atmospheric response in the middle and upper mesosphere (e.g., Pettit et al., 2019; Szelag et al., 2022)." In L144 the reference to Sinnhuber et al 2022 is correct, as comparisons of multi-model mean results using ionization rate data-sets with and without using the 90° telescope show a stronger atmospheric impact when using both telescopes consistent with the ionization rates. The reference to Pettit et al. (2019) has been added there as well.

Pettit, J. M., Randall, C. E., Peck, E. D., Marsh, D. R., van de Kamp, M., Fang, X., et al (2019). Atmospheric effects of 30-keV energetic electron precipitation in the southern hemisphere winter during 2003. Journal of Geophysical Research: Space Physics, 124, 8138- 8153. https://doi.org/10.1029/2019JA026868

L225: "up to 15% ozone reduction on average, solar cycle variations about the same magnitude"

Reply: We have made this sentence easier to read by moving the indicated statements (providing quantitative estimates of observed EPP effects on ozone) to a new sentence.

—----------------------------------------------------------------------------------------------------

**Reply to Anoruo Chukwuma**

The content of the paper is very interesting and to improve CMIP. However, I suggest a more clear deficiency of CMIP6 and what will be added to the proposed CMIP7 in solar forcing modeling. The highlights could be done in <steps-by-steps>.

We would like to thank Anoruo Chukwuma for the suggestions to improve the manuscript. In the revised version we have now made clearer what are the deficiencies of the CMIP6 forcing dataset and how these deficiencies are planned to be addressed in the revised CMIP7 dataset (see also reply to referee#1).

—----------------------------------------------------------------------------------------------------

**Reply to Gavin A. Schmidt**

We would like to thank Gavin A. Schmidt for the thoughtful and constructive suggestions and comments, which will certainly help to improve our manuscript and to consolidate the solar forcing generation for CMIP7.

This is a timely discussion, and kudos to the authors for initiating it. I have three main suggestions for the team.

1) There are many files and inputs which are being created, but they are not all equally used. Perhaps the authors could assess the literature to see what was adpoted broadly and which datasets were not as well utilised. This may inform the prioritization of ongoing and future efforts.

Reply: We agree that not all data provided in previous solar forcing datasets (such as for CMIP6) were equally used. This is particularly true for energetic particle forcing-related data, as these data cannot be fed directly into coupled models without interactive chemistry.

However, we would like to point out that energetic particle forcing is a relevant contribution to polar ozone variability, which should be considered in the ozone forcing dataset used by non-interactive models. Such an ozone forcing dataset is typically produced by chemistry-climate models which would require the energetic particle forcing data as input. In addition, our experience with the CMIP6 solar forcing has shown that these datasets are used as a reference in a broad range of applications, not being restricted only to CMIP.

2) Structural uncertainty in solar forcing, past and future, is important to characterise. While not as large as the uncertainty in aerosols, this uncertainty can play a role in the attribution of past climate changes. In PMIP3, we specifically set out alternative forcings (for the TSI and SSI) that groups could use (or not) to characterise this and this was broadly done by many groups in the last1ky experiments (Schmidt et al, 2011; 2012). I would strongly suggest doing something similar for the historical experiments. Create self-consistent separate input files based on the SATIRE and NRLSSI efforts, but do not average them to produce a 'best' guess that has not been validated - and as you note in the text, had weird inconsistencies. Ideally, and this may take some time, we would want to be creating an ensemble of reconstructions from a sampling of uncertain parameters within the reconstruction process, taking into account uncertain raw data and processing choices. Then median, maximal, and minmal reconstructions could be derived in a relatively coherent way. This might not be viable for CMIP7, but this should be the medium term aim. This would also allow for greater consistency with the PMIP5(?) forcings.

Reply: We fully agree about the importance of uncertainty quantifications and our aim is to work towards this goal already for CMIP7. As you mention, a rigorous uncertainty assessment based on ensemble reconstructions may take some time, and may thus not be feasible within the tough CMIP7 time frame. Therefore, simpler approaches should also be explored. Providing original data from ingoing reconstruction models, in addition to the reference dataset, could be one possibility.

We have now included a new section 4 in the revised version, which discusses the need for uncertainty estimates.

3) With respect to compositional feedbacks. First, these are not limited to ozone; there are also stratospheric water vapor effects via photolytic reactions with $H_2O$ and ozone-mediated changes in the oxidation of $CH_4$ - which might even be more important than some of the SEP effects?

Reply: This is an interesting comment, which we shall forward to the CMIP7 Climate Foricing Task Group for further discussion.

Second, some thought might be directed towards creating a blended ozone product that uses the observed changes in ozone that are coherent with the solar cycle together with a model based interpolation. This would have the advantage of being calibrated to the observations and not be totally beholden to the assumptions in any specific GCM (or the inputs to it, such as the uncertain background spectra). Even better would be a calibrated parameterization that provides a delta(O3) field as a funciton of the TSI/SSI change that could be valid across the PMIP, DECK and ScenarioMIP experiments. Indeed, a linearized ozone parameterization that changes as a function of TSI, temperature, etc. might provide most of what would be seen in a fully functional climate-chemistry model at a fraction of the computational cost and which would be useful far beyond the solar forcing issue.

Reply: These are all very good points, which should be discussed in the context of the generation of ozone forcing fields, which is the task of a different CMIP7 climate forcing task group. We are happy to transmit these thoughts to the latter. Regarding our manuscript, such a discussion would be out of scope since we focus on the generation of the solar forcing. With (old) Section 4, our aim was to emphasize the need for consistency among the different forcing datasets.

—-------------------------------------------------------------------------------------------------------

**Reply to Anonymous Referee #2**

We would like to thank Referee #2 for the suggestions and comments, which will certainly help to improve our manuscript and to consolidate the solar forcing generation for CMIP7.

The manuscript presents the strategy for improvement of the applied for CMIP6 simulations solar forcing data set. The authors suggested major changes inspired by the new TSIS-1 solar reference spectrum and new findings in energetic electron fluxes and spectrum description. The authors also propose switching from deterministic to stochastic ensemble forcing scenarios. In general, the manuscript is useful, but not mature enough and requires major improvements.

Major issues

1. The authors did not consider "the latest scientific advances made in the understanding of climate response" declared in the introduction

Reply: Since our manuscript is not a review paper, it was not our intention to provide an exhaustive summary of recent scientific advances made in the understanding of the climate response. Rather, our objective is to identify needs for improvement of solar forcing which can be linked to advances made in the understanding of present and past solar forcing and its impact on the climate. To make this clearer, we have changed the corresponding statement to "This paper aims to identifying improvements based on the latest scientific advances made in the reconstruction of solar forcing and in the understanding of climate response, while also 2) addressing the issues that were raised during CMIP6, and 3) facilitating the practical implementation of these data-sets, both in terms of their production and their exploitation by end users."

2. Critical analysis of the CMIP6 solar irradiation data set is missing. Some analysis in lines 26-27 is confusing. How the climate simulations were impacted, why tuning was necessary.

Reply: See also reply to Referee #1. We refer to the CMIP6 versus CMIP5 SSI differences as shown in Fig. 7 of Matthes et al. (2017). Such differences were shown to produce changes in stratospheric heating rates of up to 0.4 K/day and associated temperature changes of up to 1.5K (see Fig. 6 of Matthes et al., 2017). This information has been added.

It would be very helpful to mention how the improvement in CMIP6 mentioned in lines 22-25 were used by CMIP6 modeling teams.

Reply: These improvements were incorporated in the CMIP6 dataset to enable the use of a consistent solar forcing in a large variety of climate models, within CMIP and also beyond. These include chemistry climate models with different upper lids, some of them extending up into the thermosphere, thus requiring inputs which are typically not used by 'standard' CMIP models. To our understanding, the provision of one comprehensive and self-consistent forcing

dataset, suitable for any kind of climate model, is very useful to ensure consistency and comparability between climate simulations performed with different model types. Therefore, we would like to follow the same approach for CMIP7.

3. Any description of the requirements from CMIP7 is missing. To understand the process, it is necessary to have good understanding of the final users. For example, is it really necessary to provide extensive energetic particle forcing?

Reply: To our knowledge, no formal requirements for the generation of forcing datasets for CMIP7 have been set up. Regarding the provision of energetic particle forcing data, we would like to point out that energetic particle forcing is a relevant contribution to polar ozone variability which should be considered in the ozone forcing dataset used by non-interactive models. Such an ozone forcing dataset is typically produced by chemistry-climate models, which would require the energetic particle forcing data as input. In addition, our experience with the CMIP6 solar forcing has shown that these datasets are used as a reference in a broad range of applications, not being restricted only to CMIP.

4. The climate community is not represented in the co-author list. It is mentioned in the manuscript, that the strategy for the solar forcing definition should be developed in close collaboration with climate community. But the publication of this manuscript looks like the authors would like to have substantial help from the people representing climate community without involving them to the process.

Reply: We do not share this concern. A major fraction of co-authors of this manuscript is either involved in climate model development or in climate data analysis, apart from having expertise in solar forcing generation. Our work was initiated in the framework of SPARC's SOLARISHEPPA activity, whose overarching objective is to better understand the solar influence on climate. In addition, the purpose of this paper is to communicate our plans for CMIP7 solar forcing generation to the broader (climate modelling) community, in order to seek for feedback and constructive suggestions.

5. The plan and timeline for the development of solar forcing (including the ozone fields)for the simulations of future climate is not clearly presented.

Reply: As our manuscript focuses on the historical solar forcing, the timelines for the generation of the future solar forcing and the ozone fields are not discussed. A more general overview of the timeline for CMIP7 forcing generation can be found in Durack et al. (2023). We have included the latter reference in the revised manuscript.

Durack, P., Naik, V., Aubry, T., Chini, L., Fasullo, J., Fiedler, S., Funke, B., Graven, H., Hegglin, M., Lutron, T., MacIntosh, C., Nicholls, Z.,Plummer, D., Riahi, K., Smith, S., van Marle, M., Ziehn, T., and O'Rourke, E.: CMIP forcing datasets update timeline, https://doi.org/10.3405281/zenodo.8328527, 2023.

What is CMIP7 Climate Forcing Task Team?

Reply: See also response to Referee #1. The World Climate Research Programme (WCRP) Earth System Modelling and Observations (ESMO) project, through its Working Group on Coupled Models' (WGCM) CMIP panel and WGCM infrastructure panel (WIP), has established a number of Task Teams to support the design, scope, and definition of the next phase of CMIP and evolution of CMIP infrastructure and future operationalisation. One of these Task Teams is the Climate Forcing Task Team (https://wcrp-cmip.org/cmip7-task-teams/forcings/, see also CMIP

Annual report 20202-2023, doi:10.5281/zenodo.8101810), whose core goals is, among others, to work with teams (as ours) to identify, develop, document and deliver an updated and expanded forcing collection to near real time for CMIP7.

We have added the following sentence in the Introduction: "Note that the development and documentation of updated and expanded climate forcings for CMIP7, including the solar forcing discussed here, is coordinated by the CMIP7 Climate Forcing Task Team (wcrp-cmip.org/cmip7-task-teams/forcings/) established by the Working Group on Coupled Models' infrastructure and CMIP panels of the World Climate Research Programme's Earth System Modelling and Observations (ESMO) project."

Minor Issues

Lines 41-44: These arguments are obvious.

Reply:  We agree. Nevertheless, we think that this information might be helpful for the broader audience.

Line 43: less than 1% → around 0.1%

Reply: Agreed. We  have changed "less than 1%" to "around 0.1%".

Lines 65-68: Some illustration would be helpful. How re-normalization is performed?

Reply: Discussing and understanding these spectral differences is a vast question that goes well beyond the scope of this article. Nevertheless, we have added a simple plot.

To compare the different spectral irradiances in a meaningful way, we normalize them by forcing their TSI (= their integral) to be the same.

Line 98: Pragmatic solution is not necessary the best.

Reply: The making of such a composite data set is a multi-constrained problem, requiring high accuracy inputs but also a good understanding of the uncertainties associated with each input, the possibility to rapidly update the dataset when new data come in without affecting past observations, etc. By pragmatic we mean that it is pointless to optimize one while ignoring the others. In addition, this needs to be done with limited time resources.

Line 105: Which new data? All listed in the para or some of them?

Reply: We are specifically referring to the full-disk resolved solar images taken in the Ca II K line. This has been clarified in the revised version.

Line 108: There were alternatives to the SATIRE in PMIP4.

Reply: Yes, however, SATIRE-M was the recommended dataset.

Lines 109-115: This para is confusing. Does the future forcing will be repletion of some particular time in the past? Which time then? Yearly update for CMIP7 is strange. Do the authors really expect yearly rerun of all simulations?

Reply: See also reply to Referee#1. The usage of CMIP forcing datasets in the past was not restricted to CMIP model simulations and a broad application range is also expected for CMIP7. In particular, regularly updated forcings would be particularly beneficial for annual to decadal climate prediction as envisaged e.g. in the frame of WCRP's Explaining and Predicting Earth System Change (EPESC) Lighthouse activity. A user demand for regularly updated forcings has been identified and ways forward are currently discussed within CMIP7 Climate Forcing Task Team, see doi:10.5281/zenodo.8046147.

Lines 115-122: I understand some political background of the proposed arithmetic mean, but this approach is not scientifically solid. Potentially, the authors can establish the accuracy of both data sets relative to the observed SSI/TSI and simply apply better (even slightly better) model.

Reply: The statistical solution to this problem would be simple if we had access to comparable confidence intervals for both models. Unfortunately, no such information is available and as of today we are lacking objective reasons to give preference to one of the two models. Therefore, for the time being, we recommend to do arithmetic averaging, which is the least-committing solution.

Lines 136-137: Which minor updates are suggested?

Reply: Minor updates refers to those discussed in Section 3.3. In order to make this clearer we have changed this sentence to 'Aside from this, only minor updates with respect to M17, discussed in Section 3.3, are proposed for CMIP7 energetic particle forcing.'

Lines 139-149: Nesse Tyssøy et al., (2022) found large differences between different data sets? Why Asikainen (2019) was chosen? Maybe it is better to use arithmetic mean as suggested for SSI?

Reply: We are suggesting to apply the method presented in Nesse Tyssøy et al. (2016) on the long homogenized time series presented in Asikainen et al. (2019). This optimized loss cone estimate is not one of the ionization rates presented in Nesse Tyssøy et al. (2022), but it combines the strength of both the BCSS-LC and the Oulu estimate. It has the advantage of spanning four solar cycles applying a physics-based method to achieve a realistic estimation of the medium energy electrons precipitating into the atmosphere. An arithmetic mean, however, would cause systematic biases e.g. as function of geomagnetic latitude as the pointing direction of the two telescopes vary along its orbit.

Line 153: Now it is suggested to apply van de Kamp et al. (2016). How about strong underestimation and Asikainen (2019) mentioned in the previous section?

Reply: We agree that our statement is somewhat misleading. We have changed it to "Consistent with M17, we propose following the theoretical framework of van de Kamp et al. (2016) for parameterising the fluxes on L-shells in terms of geomagnetic index, however based on estimated electron fluxes using data from both MEPED/POES telescopes (see above)."

Lines 156-165: The proposed plan does not look feasible taking into account December 2023 deadline.

Reply: We agree that the timeline for the historical forcing generation is very demanding. Therefore, a relaxation of the timeline (until March 2024) has been agreed. The text has been changed accordingly.

Lines 191-194: Switch from dipole to IGRF could generate some jumps.

Reply: The original text was confusing. In fact, IGRF Version 12 was already recommended for CMIP6, however, only a subset of Gauss coefficients (the first 8 coefficients, corresponding to a tilted dipole approximation) were used. Thus, no jumps are expected when extending the time range with IGRF Version 13. We have replaces the text by: "For CMIP7, it is proposed to follow the approach implemented by M17 using the International Geomagnetic Reference Field (IGRF) model truncated to the eccentric tilted dipole component (the first 8 Gaussian coefficients)  which is known to adequately globally represent the realistic field for the cosmic-ray shielding (Nevalainen et al., 2013). The newest version of the IGRF, the thirteenth generation (Alken et al., 2021) model is recommended to be used."

—----------------------------------------------------------------------------------------------------

**Reply to Valentina Zharkova**

We would like to thank Prof. Valentina Zharkova for the suggestions to improve our manuscript.

Your solar forcing term is too simplified and not correct. You ver-averaging the data taking one measurement f TSI per year, In statistics averaging wrks only for the data with normal distribution. Wile variation off YSI per month and per year are far from normal.

I suggest that you should consider the ISI variations along the Earth orbit, which is shown too change significantly in this millennium as I shown in the book chapter https://www.intechopen.com/chapters/75534 and in other papers shown in my web page https://soolargsm.com, e.g.https://arxiv.org/pdf/2008.00439.pdf, https://www.scirp.org/journal/paperinformation.aspx?paperid=124007.

This will allow your codes t reflect the extra heating the Earth atmosphere gets every month during March-July every year in the millennium 1600-2600.

Reply: Please bear in mind that the CMIP7 solar forcing dataset is planned to be provided with daily and monthly resolution (not annual resolution), similar to what has been done for CMIP6. By lack of comparable confidence intervals for the different SSI datasets, or for different time resolutions (e.g., annual vs daily), unfortunately we are not in a position to consider higher order effects.

Please note that TOA solar irradiance data is provided at a fixed distance from the Sun (1 AU). Orbital variations are implicitly considered in the radiation codes of the climate models and are excluded in our SSI models.

—----------------------------------------------------------------------------------------------------

**Reply to Tom Woods (Referee #3)**

We would like to thank Tom Woods for his positive evaluation of the manuscript and constructive suggestions, which will certainly help to improve our manuscript.

This is an excellent paper summarizing the plans for updating the inputs of solar irradiance, energetic electrons, and ozone forcing data for CMIP7.  This manuscript does not provide results of those updates though, but instead is a very useful progress report on this research activity. I only have one minor (editorial) comment.

Line 108: Change "one used for for CMIP6" to "one used for CMIP6".

Reply: Thank you for spotting this typo. This has been corrected in the revised version.

—-------------------------------------------------------------------------------------------------------------

**Reply to Gareth Jones**

We would like to thank Gareth Jones for the thoughtful and constructive suggestions and comments, which will certainly help to improve our manuscript and to consolidate the solar forcing generation for CMIP7.

It is very interesting to see a paper describing how the process to define solar forcing for a future phase of CMIP will be done. It seems that the main areas of interest to those running climate simulations with solar forcings are covered. I have some views that the authors might like to consider.

* General

After reading the manuscript it is not clear to me how decisions are going to be made about what to do for CMIP7 and by who. Will it just involve the authors of this study, or a panel of some kind? Is the wider community going to be surveyed or is this paper the only route to feedback to the decision makers, via this discussion area?

Reply: The construction of the envisaged CMIP7 solar forcing dataset is endorsed by the CMIP7 Climate forcing task team (https://wcrp-cmip.org/cmip7-task-teams/forcings/) of which the first author of this manuscript is a member. This task team is one of the teams established by the World Climate Research Programme (WCRP) Earth System Modelling and Observations (ESMO) project, through its Working Group on Coupled Models' (WGCM) CMIP panel and WGCM infrastructure panel (WIP), to support the design, scope, and definition of the next phase of CMIP and evolution of CMIP infrastructure and future operationalization (see also CMIP Annual report 20202-2023, doi:10.5281/zenodo.8101810). The core goal of the Climate Forcing Task Team is, among others, to work with teams (as ours) to identify, develop, document and deliver an updated and expanded forcing collection to near real time for CMIP7. A link to this activity has been provided in the revised manuscript.

Regarding the feedback of the wider community, obtaining such feedback via the public discussion of our manuscript was a particular objective for its submission.

* Reference spectrum (Section 2.1)

It would be really helpful to include a plot showing the proposed reference spectra relative to the alternatives and past ones. This would help to understand what possible climatic impact this may have, as referred to in Lines 67-69, and as supporting evidence for the "problem" when averaging the two solar irradiance models as referred to in Lines 93-94.

Reply: We have added a figure with SSI integrated over relevant wavelength bins from the proposed and alternative reference spectra.

* Solar datasets for simulations of the past

It would also be really helpful to include a plot of the historical timeseries of the different TSI datasets and the SSI changes being described in Section 2.2, including alternative datasets not currently mentioned. Figure 1b in Yeo GRL 2020 [5] is a good example of what I had in mind.

Reply: We have included a paragraph discussing alternative datasets and a reference to the Figure 1b in Yeo GRL 2020.

Lines 97-105 & 116-121 The authors propose to just use two models of historical solar irradiance. But what about others that are available, some of which have been mentioned by the IPCC [1]. (e.g., [2,3,4])? There may be good reasons to not use or assess them - for instance the evidence for relatively large increase in TSI since the Maunder Minimum might be lacking [5] - but it would be helpful to briefly explain why the authors are excluding some datasets.

Reply: There are indeed more models on the market. More than a dozen, in fact. However, we decided to give higher priority to those models that 1) have been published in the literature and validated/tested by various users (not just the model team) and 2) whose providers are committed to updating their SSI data with more recent observations. Using these criteria, we ended up with only two candidates, namely SATIRE and NRLSSI. We agree that it would be desirable to have more independent candidates, as this would allow us to better investigate ensemble differences.

Lines 109-110 "As for CMIP6, we are planning to provide an ensemble of forcing scenarios with daily values up to 2300". The datasets provided by Matthes et al. 2017 [6] for input4MIPs were just two scenarios. A reference ('REF') and a deep minimum future ('EXT'). Not sure that could be called 'an ensemble'.

Reply: Agreed. "ensemble" has been replaced by "set".

Lines 116-121 Later on there is the discussion about future forcing uncertainty, but nothing is mentioned about what could be done for assessing historical forcing uncertainty. It may be impractical for modeling centres to run many multiple historical simulations to sample forcing uncertainties, but that is where simple models may be useful [7]. I strongly suggest that the authors also consider providing some ways of sampling the solar historical forcing uncertainty, similar to what they propose for the future scenarios. This would be in addition to the single recommended historical TSI/SSI for use by coupled models.

Reply: See also our reply to a similar comment raised by Gavin Schmidt. We fully agree about the importance of uncertainty quantifications and our aim is to work towards this goal already for CMIP7. As you mention, a rigorous uncertainty assessment based on ensemble reconstructions is challenging, and may thus not be feasible within the tough CMIP7 time frame. We have now included a discussion about the need for uncertainty estimates (new Section                                                                                                                           4).

* Solar datasets for simulations of the future

L239-L241 There are non insignificant differences between what solar irradiance models and observations give for recent TSI [9], with much larger uncertainties over longer timescales [5]. The reduction in TSI over the 21st century in the projection [6] should probably be mentioned. Is it too early to assess it? My opinion is that it was too speculative to propose for use with climate models.

Reply: This reduction in CMIP6 is a plausible scenario, thus of course speculative. However, we have produced an extended historical CMIP6 dataset up to 2020 for the SPARC's  CCMI-2022

model intercomparson exercise (see https://solarisheppa.geomar.de/solarisheppa/ccmi2022), which shows that the CMIP6 projection for the 2015-2020 period was even sub-estimating the reduction of the solar activity level, in particular during the solar cycle minimum around 2019. See Figure 1 below.

[Figure]

**Figure 1: Time evolution of TSI, integrated SSI in 200-400 nm and 400-700 nm bands, as well as mid-energy electron (MEE) ionization rate at 0.001 hPa form the CMIP6 forcing data set (dashed) and the CCMI-2022 forcing dataset (solid). The overlap region of both datasets is 2015-2020.**

I think rather than saying "realistic" for the Matthes et al. 2017 [6] dataset projection, a better word is 'plausible'.

Reply: Agreed, We have changed "realistic" to "plausible".

L251-252 I suggest expanding on what is meant by "stochastic ensemble forcing scenarios". Does this suggest a large ensemble of TSI timeseries, say, each corresponding to different plausible evolutions of solar cycle lengths and amplitudes and longer time scale magnitude?

Reply: Exactly, our suggestion is to produce an ensemble of forcing datasets (including all radiative and particle forcing components in a self-consistent manner) which are constructed from different plausible evolutions of the solar activity level, i.e., considering different solar cycle lengths, amplitudes, and distribution of impulsive events like SPEs. We now state: "Regarding the future solar forcing, such an ensemble could be constructed from a set of plausible evolutions of the solar activity level, i.e., considering different solar cycle lengths, amplitudes, and distribution of impulsive events like solar proton events."

Expecting modeling centres to use multiple future solar irradiance datasets in their coupled models may be optimistic to say the least. The effort to create ancillary files to run on each

model, for each "ensemble forcing" should not be underestimated. Institutions submitted ssp245 (as an example) simulations for 46 models. Most of them (~30) submitted initial condition ensembles of three or less, while only 13 submitted greater than 5 ensemble members. Thus it is likely only a few models will be able to sample some of the proposed future solar forcing uncertainty. A danger could be that some of those solar forcing ensembles are over sampled by those models with few ensemble members, reducing their usefulness. It is likely that only a few modeling centres will have the resources to submit a reasonable number of simulations that sample the solar forcing ensemble.

Reply: We agree that the use of an ensemble future solar forcing would be extremely demanding from a computational point of view. At this point in time, our intention was limited to initiate a discussion in a broader community, which, in turn, might trigger some further thinking and conceptual work, and eventually may result in a dedicated MIP activity. One possibility to avoid unrealistic computational overload could be the application of statistical emulators, based on DA methods, rather than using full climate model simulations, to explore the ensemble forcing. In any case, in order to avoid problems caused by oversampling of individual forcing ensemble members, our intention is to provide one single reference future forcing for DECK and MIP activities not dedicated to the assessment of projected natural forcing uncertainties.

Matthes 2017 [6] provided two future TSI/SSI scenarios ('REF' and 'EXT'). I have come across only one study that has used the 'EXT' scenario, and that study concluded that "low amplitude solar forcings such as the EXT CMIP6 or similar are not worthwhile considering during the next CMIP type of activities." [8] (are the authors aware of this reference?).

Reply: Yes, we are aware of this study by Sedlacek et al., which is also in line with previous assessments. However, this study (and previous studies) did not assess the impact of the uncertain solar forcing evolution on shorter scales (e.g., related to solar cycle progression and impulsive events) which might introduce important natural forcing uncertainties on the annual to decadal timescales. This impact cannot be assessed by means of CMIP6 future scenarios since both REF and EXT scenarios were based on one single projection of historical cycles into the future (however differently scaled).

While providing forcing ensembles could be useful for use in simple models [7], please consider also retaining a single recommended solar irradiance dataset for most coupled model submissions to use.

Reply: Yes, we agree that a single reference solar forcing dataset should be recommended for CMIP7 DECK and MIP simulations which are not dealing with the assessment of natural forcing uncertainties.